# Mobile Brain Imaging to Examine Task-Related Cortical Correlates of Reactive Balance: A Systematic Review

**DOI:** 10.3390/brainsci12111487

**Published:** 2022-11-02

**Authors:** Rudri Purohit, Tanvi Bhatt

**Affiliations:** 1Department of Physical Therapy, University of Illinois at Chicago, Chicago, IL 60612, USA; 2Ph.D. Program in Rehabilitation Sciences, College of Applied Health Sciences, University of Illinois at Chicago, Chicago, IL 60612, USA

**Keywords:** cortex, neuroimaging, reactive balance, compensatory stepping, unpredicted perturbations

## Abstract

This systematic review examined available findings on spatial and temporal characteristics of cortical activity in response to unpredicted mechanical perturbations. Secondly, this review investigated associations between cortical activity and behavioral/biomechanical measures. Databases were searched from 1980–2021 and a total of 35 cross-sectional studies (31 EEG and 4 fNIRS) were included. Majority of EEG studies assessed perturbation-evoked potentials (PEPs), whereas other studies assessed changes in cortical frequencies. Further, fNIRS studies assessed hemodynamic changes. The PEP-N1, commonly identified at sensorimotor areas, was most examined and was influenced by context prediction, perturbation magnitude, motor adaptation and age. Other PEPs were identified at frontal, parietal and sensorimotor areas and were influenced by task position. Further, changes in cortical frequencies were observed at prefrontal, sensorimotor and parietal areas and were influenced by task difficulty. Lastly, hemodynamic changes were observed at prefrontal and frontal areas and were influenced by task prediction. Limited studies reported associations between cortical and behavioral outcomes. This review provided evidence regarding the involvement of cerebral cortex for sensory processing of unpredicted perturbations, error-detection of expected versus actual postural state, and planning and execution of compensatory stepping responses. There is still limited evidence examining cortical activity during reactive balance tasks in populations with high fall-risk.

## 1. Introduction

Maintaining upright posture and balance is critical for humans to carry out activities of daily living. Human balance control relies on anticipatory and reactive domains, respectively to maintain postural stability in response to self-induced/predicted perturbations and to restore stability after encountering unpredicted perturbations [1,2,3,4,5]. Both anticipatory and reactive domains involve coordination of sensorimotor and cognitive systems for maintaining upright posture or recovering from balance losses and to avoid falling [6,7,8,9,10,11]. From a neural control perspective, it is well established that the cerebral cortex modulates volitional/anticipatory postural and balance adjustments prior to goal-directed movements such as reaching or obstacle crossing [12,13,14,15,16,17]. Previously, three reviews presented findings on cortical involvement in planning and execution of anticipatory/volitional balance and gait-related tasks among healthy young, healthy older, and clinical populations [18,19,20]. However, there is limited evidence regarding the cortical mechanisms and/or substrates involved with initiation, execution, and modulation of reactive balance responses. 

Reactive balance responses elicited upon unpredicted perturbations include feet-in-place responses (i.e., hip or ankle strategies) [21] or change-in-support responses (i.e., compensatory stepping or grasping) [22] and are known to be influenced by the magnitude of the perturbation (i.e., its displacement, velocity, or acceleration) [23]. Cumulative evidence from animal and human studies indicated that feet-in-place responses might be elicited by spinal cord and brainstem mediated short and medium latency reflexes [23,24,25]. Conversely, change-in-support (i.e., compensatory stepping) strategies are postulated to be ‘triggered’ from brainstem control centers [18,23,24]. Further, behavioral/neurophysiological human studies indicated the possible role of the cerebral cortex in modulating spatiotemporal characteristics of compensatory stepping based on factors such as prior experience, task demand/difficulty and current context [14,26,27,28,29,30,31]. Studies have also indicated that physiological aging [32,33,34,35,36] and neurological impairment [37,38,39,40] are associated with alterations in compensatory stepping and impaired reactive balance control, which might be related to changes in cortical substrates. Evidence from studies using neuroimaging modalities supported these postulations from behavioral studies regarding possible cortical modulation of reactive balance responses [41,42,43,44,45,46,47,48,49,50,51]. In the past 2 decades, several neuroimaging modalities have been used by researchers to examine reactive balance control including magnetic resonance imaging (MRI) [52,53,54], functional MRI (fMRI) [55,56], positron emission tomography (PET) [57,58], electroencephalography (EEG) [41,42,59,60,61,62,63], and functional near-infrared spectroscopy (fNIRS) [50,51,64,65]. Studies have used fMRI to examine task-related changes in the cortico-subcortical areas for simulating unpredicted perturbations through mental imagery/observation of tasks [66,67]. However, with poor temporal resolution (2 s), fMRI might not provide information on true event-related activations especially for reactive balance control (e.g., at the instance a slip occurs while observing a video in the scanner) [68,69]. Alternatively, recently developed wireless EEG systems allow full subject mobility and can record brain activity during actual tasks. While EEG lacks spatial resolution (6–27 cm^3^), it offers excellent temporal resolution (up to 1 ms) and can record task-induced evoked potentials over all cortices [20,68]. fNIRS also allows full subject-mobility, however, the brain surfaces recorded might be often limited to prefrontal and frontal cortices [70]. Nevertheless, fNIRS offers greater spatial resolution (1–2 mm^3^) than EEG (6–27 cm^3^) and higher temporal resolution (up to 10 ms) than fMRI (∼3 s) over pre-frontal and frontal cortical areas [70,71]. Moreover, both EEG and fNIRS are portable, wireless and can be good tools to examine cortical activations in clinical and research settings [20,68]. 

The advantages of EEG have motivated researchers to use this technique for examining cortical evoked potentials associated with reactive balance control. There are two recently published reviews examining perturbation-evoked potentials (PEPs) recorded via EEG [69,72]. PEPs are multicomponent cortical responses comprised of positive and negative potentials recorded within the first 400 ms following an external perturbation. Payne et al. [72] integrated evidence on association between error-related negative cortical potentials recorded during volitional (stimulus–response) and reactive balance tasks, whereas Varghese et al. [69] focused on examining characteristics of PEPs in response to sensory (visual/auditory/proprioceptive) and mechanical perturbations. However, both Payne et al. [72] and Varghese et al. [69] did not provide detailed evidence on event-related changes in frequency spectrums following unpredicted perturbations. Event-related spectral perturbations (ERSPs) are one type of stimulus or task-related changes in cortical frequencies recorded via EEG and assessed via time-frequency analysis of brain activity. Further, no reviews have included findings on hemodynamic changes during unpredictable perturbations recorded via fNIRS. Lastly, there is still lack of evidence regarding the associations between unpredicted perturbation-induced cortical and behavioral/biomechanical responses. Therefore, this systematic review aimed to examine evidence on cortical activations using mobile neuroimaging techniques (EEG and/or fNIRS) during different types of reactive balance tasks among healthy young, older, and neurologically impaired populations. Additionally, this review aimed to examine the current evidence that exists regarding associations between cortical and behavioral or biomechanical measures of balance and gait.

## 2. Methods

### 2.1. Review Objectives

This systematic review examined literature on neuroimaging of reactive balance control from 1980–2021 to address the following research questions: 

**Research question (RQ1):** What are the temporal and spatial characteristics of cortical activity recorded via EEG and fNIRS during novel unpredicted mechanical perturbations delivered across different tasks? 

**Research question (RQ2):** What are the temporal and spatial characteristics of cortical activity recorded via EEG and fNIRS during repeated unpredicted mechanical perturbations delivered across different tasks? 

**Research question (RQ3):** Are behavioral/biomechanical outcomes correlated with cortical activity during unpredicted mechanical perturbations delivered across different tasks?

### 2.2. Review Methodology

#### 2.2.1. Search Process and Strategy

To address the proposed research questions, we used four databases: PubMed, Web of science, IEEE explore and Google scholar. The database search key terms were found in one of the following sections of a research article: Title, Abstract or Keywords. Following the initial *advanced* search, the results from each database were run through Mendeley to detect any duplicate studies. Once all the duplicate studies were removed, the results from the databases were uploaded to Rayyan—Intelligent systematic review software [73]. Each of the authors independently included and excluded research studies while being blinded to other’s decision. 

We used the Preferred Items for Systematic Reviews and Meta-Analyses method (PRISMA: see Figure 1) [74]. The search strategy included combinations of terms related to neuroimaging studies that incorporated reactive balance paradigms. The search fields were connected with “AND” to ensure that matches included key terms from both search fields and within each search field, terms were connected with an “OR” so that at least one of the key terms was matched in the results. The advanced search terms for neuroimaging included variations of the following terms: electroencephalography, EEG, functional near-infrared spectroscopy, fNIRS, alpha, beta, gamma, delta, perturbation-evoked potentials, PEP, N1, P1, N2, and P2, cortical activity, neural activity, and brain activity. The advanced search terms for reactive balance control paradigms included variations of all the following terms: reactive balance, perturbed gait, perturbed stance, unpredicted perturbations, slips, trips, dual task slips, dual task trips, and falls. An example from the search combinations is listed below: 

((EEG and reactive balance)) OR ((EEG and reactive balance and perturbations)) OR ((EEG and slips)) OR ((EEG and trips)) OR ((EEG and perturbed gait)) OR ((EEG and dual-task and reactive balance)) OR ((EEG and dual task and perturbations)) OR ((EEG and N1 and reactive balance)) OR ((EEG and theta and reactive balance)) OR ((EEG and beta and reactive balance)) OR ((EEG and gamma and reactive balance)) OR ((EEG and alpha and reactive balance) ((EEG and N1 and perturbations)) OR ((EEG and theta and perturbations)) OR ((EEG and beta and perturbations)) OR ((EEG and gamma and perturbations)) OR ((EEG and alpha and perturbations)).

#### 2.2.2. Eligibility Criteria

The inclusion and exclusion criteria for this review were based on neuroimaging studies published from 1980–2021 that examined cortical changes during reactive balance tasks (via EEG and fNIRS). Studies were included based on the following criteria: (1) studies that delivered single or repeated unpredicted mechanical perturbations across one or more functional task(s) including sitting, standing, and walking, (2) studies that recorded brain activity only via EEG and fNIRS, and (3) research articles that were in English. Studies were excluded based on the following criteria: (1) studies that assessed static or dynamic volitional balance control (e.g., avoiding obstacles while walking or catching a ball), (2) studies that investigated the effects of training, exercise intervention, drug or therapy, (3) studies that did not assess cortical activity during the actual task (i.e., studies that assessed cortical activity before or after reactive balance tasks via fMRI, single-photon emission computerized tomography-SPECT, PET, or diffusion tensor imaging), (4) studies that delivered only sensory perturbations (i.e., visual, auditory or somatosensory perturbations), and (5) animal neuroimaging studies. 

#### 2.2.3. Quality Assessment

The quality of included studies was assessed using the fourteen-point National Institute of Health (NIH) Study Quality Assessment for Cross-sectional studies. Previous systematic reviews have used this tool to assess study quality [75]. Each of the fourteen questions was assigned a score of “2” if the criterion was met by the research paper, “1” if the criterion was partially met, “0” if the criterion was not met and N/A if the criterion was not applicable or not related to the research paper. The sum of points from all the fourteen questions divided by the total of points applicable provided a score which was then used to rate each research study (≥75% = high, 55–75% = moderate, ≤55% = low) (Table 1). Each author rated all the included studies independently. 

The majority of the studies (≈20/35) received a moderate rating from both the authors. General strengths of eligible studies included clear specification of study population(s), specification of inclusion/exclusion criteria and usage of valid and reliable tools for assessment of outcomes. Common limitations of eligible studies included lack of sample size justification, power description/analysis of variance and effect estimates. Specifically, none of the included studies provided justification for sample size for the included population(s). Secondly, a majority of studies did not incorporate key confounding factors that could impact the outcome measures into statistical analysis. Lastly, in some studies the aims/objectives/research question and the independent and dependent outcome variables were not clearly stated or defined.

#### 2.2.4. Data Extraction

Following independent evaluation of the articles, both the authors discussed and decided whether to include or exclude the research studies under “undecided/maybe section”. Both the reviewers first worked on retrieving the data independently and then together for summarizing the results tables. Specific information was extracted from all the eligible studies including (1) first author and year of publication; (2) sample size of the study; (3) sample characteristics (e.g., age, gender and population); (4) characteristics of disease/disorders (for studies that included clinical populations); (5) neuroimaging technique(s) used; (6) information about the type, position, mode, intensity of reactive balance tasks; (7) behavioral/kinematic/biomechanical outcomes assessed; (8) key findings that specifically address RQ1–RQ3; (9) study limitations. All the above information for studies in all categories and subsets is presented in Table 2, Table 3, Table 4, Table 5, Table 6 and Table 7.

#### 2.2.5. Data Analysis 

Search results obtained from four research databases yielded a total of 1022 research articles. After removing all the duplicate results via Mendeley we extracted 422 original research articles. After reviewing the abstracts of all these articles, a total of 45 articles were included. The reasons for exclusion of other 377 were (1) studies that delivered predicted mechanical or sensory perturbations for examining cortical activity, (2) studies that delivered unpredicted sensory perturbations (i.e., visual or proprioceptive perturbations, (3) intervention studies that examined the effect of treatment on changes in cortical activity to unpredicted perturbations, (4) studies that delivered unpredicted perturbations to assess changes in cortical activity in children and adolescents (below 18 years of age). The remaining 45 articles were assessed for eligibility and 10 were removed based following points: (1) studies that were not in English, (2) Animal studies, (3) Studies that delivered sensory perturbations, and (4) Studies that delivered only predicted perturbations. This was followed by full-text examination that yielded a total of 35 cross-sectional studies that assessed changes in cortical activity during novel or repeated unpredicted mechanical perturbations (Figure 1) [41,42,43,44,48,50,51,59,60,63,64,65,76,77,78,79,80,81,82,83,84,85,86,87,88,89,90,91,92,93,94,95,96,97,98].

## 3. Results

The results section was summarized based on the research questions (RQ1–3) and included studies from two primary domains: (1) studies using EEG to record cortical activity (*n* = 31) and (2) studies using fNIRS to record cortical activity (*n* = 4) during unpredicted mechanical perturbations. The EEG studies were sub-divided into 2 categories: studies examining PEPs (*n* = 20) and studies examining changes in frequency spectrums (*n* = 11). The fNIRS studies (*n* = 4) examining hemodynamic changes were summarized together. Further, each category was divided into type of reactive balance task including sitting, standing, walking and multiple tasks. Within each task, studies that assessed healthy young, heathy older and pathological populations were reported separately. Finally, for each population the results were reported based on the type of perturbations (i.e., unpredicted only, predicted versus unpredicted and dual tasks). 

Although only 12 studies examined the associations between cortical activations and behavioral/biomechanical outcomes [41,44,63,65,82,83,85,87,93,94,95,96], most of the included studies assessed behavioral/biomechanical measures. Among the included studies, behavioral measures such as dynamic balance were assessed using Berg Balance Scale [65,83,95], Mini Balance Evaluations System Test (Mini-BESTest) [85] and the distance travelled on a Beam-Walking Task [44,82]. Other balance measures were single foot stance time, functional base of support and time-to-step using sensory organization testing [63]. Mobility was assessed using Short physical performance battery to stratify older adults into mobile and frail older adults [63]. In addition, psychological factors like pain perception, coping strategies, fear of falling, perceived balance confidence and perceived stability were assessed with clinical questionnaires [41,93] and autonomic responses were assessed by examining changes in electrodermal responses [87]. Lastly, biomechanical and kinematic outcomes including center of mass/pressure displacement [93,95], postural sway [94] and electromyography (EMG) responses for lower limb muscles were assessed during reactive balance tasks [96]. Behavioral/biomechanical outcomes for only studies that examined corresponding cortical activations during reactive balance tasks are listed in the tables (Table 2, Table 3, Table 4, Table 5, Table 6 and Table 7).

### 3.1. Spatial-Temporal Characteristics of Cortical Activity: Novel Perturbations (RQ1)

#### 3.1.1. EEG Studies—PEPs

This section included 20 studies that assessed PEPs during unpredicted perturbations in sitting (*n* = 1) [76], stance (*n* = 17) [41,44,59,63,77,78,79,80,84,87,89,90,91,92,93,94,95], walking (*n* = 1) [60] and multiple tasks (*n* = 2) [42,60] (Table 2). A majority of studies included young adults (*n* = 17), some studies compared healthy and frail older adults (*n* = 1) [63], people with traumatic brain injury (*n* = 1) [95] and people with low back pain (*n* = 1) [93] with age-matched/healthy controls. Studies analyzed one or more early or late PEPs (Early PEPs: 0–150 ms post-perturbation—P1 and N1, late: 250–400 ms post-perturbation- P2 and N2) in response to unpredicted perturbations. 

Task—Sitting:

*Healthy young population: Unpredicted perturbations only:* Ditz and colleagues [76] assessed PEP characteristics during medial-lateral seated perturbations using a tilting chair. This study reported peak N1 (Amplitude-amp): −28.3 ± 14.5 µV, Latency-lat: 144 ± 9 ms post-perturbation (Mean ± SD) focused on midline central (Cz) with distribution around frontal, central and parietal areas. Peak P2 (amp: 12.2 ± 4.1 ± 4.1 µV, lat: 330 ± 30 ms) was centered at midline parietal electrode (Pz) and distributed around parietal areas. However, P1 and N2 PEPs were not identifiable using single-trial EEG analysis. 

*Healthy older population:* None of the included studies assessed PEPs during unpredicted perturbations in sitting in healthy older adults.

*Pathological populations:* None of the included studies assessed PEPs during unpredicted perturbations in sitting in pathological populations.

Task—Standing: 

*Healthy young population: Unpredicted perturbations only:* Goel et al. [78] examined the effect of movable platform speed (low: 7.9 cm/s, high: 15.88 cm/s) and direction (forward/backward) on N1 response. Larger N1 amplitudes (High-backward: −10.8 ± 1.7 µV/High-forward: −10.5 ± 1.3 µV vs. Low-backward: −8.1 ± 1.1 µV/Low-forward: −8.0 ± 0.9 µV) and shorter latency (High-backward: 159.7 ± 4.4 ms/High-forward: −159.3 ± 5.4 ms vs. Low-backward: 167.3 ± 3.4 ms/Low-forward: 171.3 ± 5.1 ms) for high-speed compared to low-speed perturbations was reported. Quant and colleagues [80] examined changes in late PEPs (P2 and N2) during immediate and delayed perturbation deceleration. No differences in early (P1 and N1) and late (P2 and N2) PEPs were observed between the two conditions. Further, Payne and Ting, 2020a and 2020b [43,44] examined the effect of backward support-surface translations delivered via movable platform on N1 characteristics over central electrodes. All participants were instructed to plan a compensatory step on half of the trials and avoid stepping on the other half [43,44]. In both studies, perturbations were delivered at three magnitudes (easy, moderate, and difficult: 8–22 cm distance). One of the studies found no differences in N1 amplitude based on the planning of compensatory steps [43]. However, this study reported significantly larger N1 amplitudes (11% or 6.3 µV higher) when compensatory steps were executed compared to those where participants resisted taking a step. The other study reported increase in N1 amplitude with increase in perturbation magnitude such that N1 amplitude increased 5–6 µV with each perturbation magnitude. This study also reported correlations between N1 amplitude and clinical balance performance. Those results have been discussed in RQ3 section. 

*Predicted and Unpredicted perturbations:* Adkin et al. [59] delivered manual unpredicted and predicted forward trunk perturbations to healthy young adults. P1, observed in less than half (3/8) of subjects, showed no differences in amplitude during predicted (Amp: ≈4.2–5.5 µV, Lat: ≈28–61 ms post-perturbation) or unpredicted (Amp: ≈5.5–6.7 µV, Lat: ≈28–61 ms post-perturbation) perturbations. Large N1 amplitudes were reported during unpredicted perturbations (≈−7.5 to −22 µV) at fronto-central and centro-parietal (FCz, Cz and CPz) electrodes. P1 and N1 during predicted perturbations were non-distinguishable from the background EEG activity. A similar study by Mochizuki et al., 2008 delivered predicted and unpredicted backward perturbations using lean-and-release of a load in standing [84]. During predicted perturbations, greater pre-perturbation activity was observed when compared to unpredicted perturbations. Additionally, N1 amplitude during unpredicted perturbations was significantly larger (32.0 ± 14.8 µV) than predicted perturbations (17.6 ± 7.2 µV). The N1 latency during unpredicted perturbations (156.5 ± 11.8 ms) was significantly increased when compared to predicted perturbations (140.1 ± 25.9 ms). Two studies (Adkin et al., 2008 and Sibley et al., 2010) [41,87] exposed young adults to predicted and unpredicted manual trunk [41] and load-release [87] perturbations at LOW (ground) and HIGH (1.6/3.2 m above ground) levels. In both studies, larger N1 amplitudes during HIGH (89.2 ± 17 µV) compared to LOW (70.2 ± 14 µV) were reported during unpredicted perturbations [41,87]. However, there were no changes in N1 latency and P2 characteristics between the two conditions [87]. Additionally, N1 was not generated during predicted perturbation but instead a negative cortical potential was observed before the onset of perturbation that showed no changes between HIGH and LOW ground perturbations. 

*Dual tasks:* Four studies examined the effect of concurrent cognitive task during mechanical perturbations in young adults [89,90,91,92]. There were no task-related differences observed in P1 in any of these studies. All studies reported significant reduction in N1 amplitude with no differences in N1 latency and P2 characteristics during dual task compared to single task (i.e., cognitive task-one study or balance tasks-three studies). However, larger N1 amplitudes during dual task compared to single task even during predicted perturbations were reported [91]. Further, Bogost and colleagues [89] identified specific regions of cortical activation during single and dual tasks. During single tasks (balance task), activations occurred primarily in the prefrontal, primary motor cortex and supplementary motor area predominantly in the left hemisphere. However, during dual task, activations was seen in prefrontal, frontal, sensorimotor and parietal cortices predominantly in the right hemisphere. 

*Healthy older population: Unpredicted perturbations only:* Duckrow et al., 1999 compared PEPs during unpredicted stance perturbations in healthy and frail older adults with healthy young adults. Delayed P1 latency (Old: 87 ± 8 ms; Young: 60 ± 5 ms) and smaller N1 amplitudes (Old: 23 ± 12, Young: 45 ± 17 µV) were seen in older (mobile and frail) compared to young adults [63]. 

*Pathological populations: Unpredicted perturbations only:* Two studies compared PEPs during stance perturbations in people with traumatic brain injury and low back pain with their younger or age-matched controls [93,95]. In people with chronic low back pain, no differences in N1 and larger P2 amplitudes at fronto-central areas (FCz and Cz) compared to age-matched healthy controls were reported [93]. On the other hand, smaller N1 amplitudes (2.1 ± 1.3 μV) with no differences in N1 latency were reported in people with chronic traumatic brain injury when compared to healthy controls (3.8 ± 1.2 μV). 

Task-Walking: 

*Healthy young population: Unpredicted perturbations only:* One study compared changes in cortical activity during unpredicted gait perturbations with stance perturbations [60]. This study has been mentioned in the multiple task section below. 

*Healthy older population:* None of the included studies assessed PEPs during unpredicted perturbations in walking in healthy older adults.

*Pathological populations:* None of the included studies assessed PEPs during unpredicted perturbations in walking in pathological populations.

Multiple tasks:

*Healthy young population: Unpredicted perturbations only:* Two studies compared PEPs during multiple tasks [42,60]. Dietz et al. [60] compared changes in cortical activity during unpredicted gait perturbations with stance perturbations delivered via motorized treadmill. P1 during gait perturbations exhibited longer latency (80 ± 11 ms) compared to stance perturbations (43 ± 6 ms) at midline central area (Cz). In addition, N1 observed during gait perturbations exhibited longer latency (131 ± 7 ms) compared to stance perturbations (95 ± 8 ms). In addition, Mochizuki et al. [42] compared changes in PEPs characteristics during seated and stance perturbations. Larger P2 amplitude with no changes in latency during stance (37.87 ± 6.14 μV) compared to sitting tasks (23.66 ± 6.21 μV) was observed at centro-parietal areas (CPz) [42]. 

*Healthy older population:* None of the included studies assessed PEPs during unpredicted perturbations in one or more tasks in healthy older adults.

*Pathological populations:* None of the included studies assessed PEPs during unpredicted perturbations in one or more tasks in pathological populations.

#### 3.1.2. EEG Studies—Changes in Frequency Spectrums

This section included 8 studies that assessed changes in low-to-high frequency bands in response to unpredicted perturbations in stance (*n* = 5) [49,77,79,81,82,83,85], walking (*n* = 2) [86,88] and multiple tasks (*n* = 1) [48] (stance and walking) (Table 3). While a majority of these studies included young adults (*n*= 5), one study examined cortical activations in healthy older adults [85] and two studies compared cortical activations in people with chronic cortical stroke [49] and chronic traumatic brain injury [83] with healthy counterparts. Studies analyzed changes in cortical frequencies including alpha (7.5–12.5 Hz), beta (13–30 Hz), gamma (40–100 Hz) and theta (4–8 Hz). For simplicity, we would like to clarify the meaning of terms related to frequency analysis: synchronization or enhancement refers to increase in frequency band(s) compared to baseline whereas desynchronization or suppression refers to decrease in frequency band(s) compared to baseline [81]. Event-related spectral perturbations (ERSPs) measure the average changes in spectral power over varied periods of time [77] and spectral power density is measured in terms of absolute spectral power density or average spectral power density. 

Task—sitting: None of the included studies assessed changes in cortical frequencies during seated perturbations in healthy young, healthy older or pathological populations. 

Task—Standing: 

*Heathy young population: Unpredicted perturbations only:* Preliminary studies examined the relationship between frequency characteristics and PEPs in young adults during unpredicted forward perturbations via lean-and-release method [77,79]. They reported corresponding increases in alpha, beta and theta frequencies in fronto-central and parietal areas during the time of N1 occurrence [79]. Further, Solis-Escalante et al. [81] exposed young adults to unpredicted backward perturbations with random visual cues to either execute in-place or compensatory step. Reduction in theta and alpha bands was observed during observation of cues followed by enhancement during response preparation phase (5.6–7 s after cue). However, the reduction in alpha bands for feet-in-place responses during cue observation was maintained throughout the response preparation phase. On the other hand, both feet-in-place and compensatory stepping shared some common frequency characteristics including initial alpha band suppression in midline occipital areas during cue observation followed by beta I and II reduction in midline central and parietal area, respectively during response preparation phase (5.6–7 s after cue observation). Additionally, Ghosn et al. [82] exposed young adults to unpredicted backward perturbations at different intensities (i.e., acceleration and displacement) via a movable platform. This study reported increase in beta power in central areas 50 ms post-perturbation especially during perturbations with higher intensity. 

*Healthy older population:* None of the included studies assessed changes in frequency spectrums during unpredicted perturbations in standing in healthy older adults.

*Pathological populations: Unpredicted perturbations only:* Solis-Escalante et al. [49] exposed people with chronic stroke and young adults to unpredicted stance perturbations via movable platform. This study reported perturbation direction-specific pictographic presentations of theta band activation patterns similar in both groups. Such direction-specific cortical changes in theta bands were accompanied by corresponding scalp topographies with frontal, central and parietal areas. Shenoy et al., 2021 exposed people with chronic traumatic brain injury and age-matched healthy adults to unpredicted forward/backward variable magnitude perturbations and compared network connectivity strength in alpha, beta and theta bands [83]. Network strength analysis showed task-related reductions in theta and alpha with no differences in beta network strength in people with traumatic brain injury compared to healthy counterparts. Lastly, Palmer et al., 2021 delivered multi-directional perturbations via movable platform to healthy older adults and reported correlations between beta connectivity and balance performance measures [85]. The study results are explained in RQ3 section. 

Task—Walking: 

*Healthy young population: Unpredicted perturbations only:* Two studies assessed band power [86] and rate of variation of spectral power density [88] in frequency bands in young adults during forward [88] and backward perturbations [86] in walking using a split-belt treadmill. During post-perturbation recovery period [86], reduction in alpha power was observed in sensorimotor and posterior-parietal areas compared to unperturbed periods. Additionally, theta power increased during walking in posterior parietal cortex and in sensorimotor and posterior parietal cortices during recovery compared to unperturbed periods. On the other hand, post-perturbation theta and beta frequency increased with highest difference in beta band over fronto-central and parietal electrodes [88]. 

*Healthy older population:* None of the included studies assessed changes in frequency spectrums during unpredicted perturbations in walking in healthy older adults.

*Pathological populations:* None of the studies included assessed changes in frequency spectrums during unpredicted perturbations in walking in pathological populations.

Multiple tasks: 

*Healthy young population: Unpredicted perturbations only:* A study conducted by Peterson and Ferris [48] examined the differences in frequency characteristics during medio-lateral stance and gait perturbations via manual waist pulls. Increased alpha power and beta power in sensorimotor and increased alpha power in supplementary motor and posterior parietal cortices were observed during stance compared to walk perturbations. Theta synchronization in supplementary motor area and anterior cingulate areas were observed during perturbation onset in both conditions. This was followed by alpha-beta desynchronization in bilateral sensorimotor areas during stance and walk perturbations. 

*Healthy older population:* None of the included studies assessed changes in frequency spectrums during unpredicted perturbations in more than one task in healthy older adults.

*Pathological populations:* None of the included studies assessed changes in frequency spectrums during unpredicted perturbations in more than one task in pathological populations.

#### 3.1.3. fNIRS Studies—Changes in Hemodynamic Responses

This section included 4 fNIRS studies that assessed changes in hemodynamic responses during stance (*n* = 2) [64,65] and gait perturbations (*n* = 2) [50,51] (Table 4). Three studies assessed healthy young populations whereas 1 study included adults with subcortical stroke [65]. 

Task—Sitting: None of the included studies assessed changes in hemodynamic responses during seated perturbations in healthy young, healthy older, or pathological populations.

Task—Standing:

*Healthy young population: Unpredicted perturbations only:* None of the included studies assessed changes in oxygenated hemoglobin only during unpredicted stance perturbations in healthy young adults. 

*Predicted and unpredicted perturbations:* A study in healthy young adults [64] reported changes in oxygenated hemoglobin based on the perturbation type. While both predicted (with auditory signals) and unpredicted (without auditory signals) perturbations resulted in increase in oxygenated hemoglobin in frontal and parietal areas, predicted perturbations resulted in greater increase in oxygenated hemoglobin in right superior parietal lobes and left supplementary motor area as well as activation of left middle frontal gyrus and precentral gyrus. Although majority of temporal characteristics were similar between the two conditions, the right middle frontal gyrus displayed early increase in oxygenated hemoglobin in predicted compared to unpredicted perturbations. 

*Healthy older population:* None of the included studies assessed changes in hemodynamic responses during unpredicted perturbations in standing in healthy older adults.

*Pathological populations: Unpredicted perturbations only:* A study conducted in people with subcortical stroke delivered anterior-posterior stance perturbations via a movable platform. This study reported increased oxygenated hemoglobin in bilateral prefrontal cortices and premotor cortex and parietal association area of the unaffected hemisphere [65]. 

Task—Walking:

*Healthy young population: Unpredicted perturbations only:* Koren et al., 2019 exposed healthy young adults to unpredicted overground perturbations during walking via customized shoe-like mechanical system [50]. Increased oxygenated hemoglobin was reported in bilateral prefrontal cortices during perturbed compared to unperturbed walking/standing periods. 

*Healthy older population:* None of the included studies assessed changes in hemodynamic responses during unpredicted perturbations in walking in healthy older adults.

*Pathological populations:* None of the included studies assessed changes in hemodynamic responses during unpredicted perturbations in walking in pathological populations.

### 3.2. Spatial-Temporal Characteristics of Cortical Activity: Repeated Perturbations (RQ2)

#### 3.2.1. EEG Studies: PEPs

Two EEG studies assessed changes in PEPs during repeated stance perturbations in healthy young population [94,96] (Table 5). 

Task—Sitting: None of the included studies assessed changes in PEPs during repeated seated perturbations in healthy young, healthy older and pathological populations. 

Task—Standing: 

*Healthy young population:* Unpredicted perturbations only: Studies by a group of researchers [96] tested the effect of repeated backward platform perturbations by delivering easy (displacement 8 cm), moderate (13–15 cm) and difficult (18–22 cm) perturbations with variable acceleration levels. Young adults were instructed to plan a stepping response in half of the trials and to resist stepping in the other half [43]. The results showed an increase in N1 amplitude and reduction in N1 latency with increasing perturbation difficulty (difficult > moderate > easy) [72]. Additionally, repeated perturbations resulted in reduction in N1 amplitude [72]. Another study by Mierau et al., 2015 [94] tested the effect of repeated perturbations on N1 amplitude while balancing on their dominant leg. The results showed no differences in P1 characteristics between the trials. However, the study reported reduction in N1 amplitude between novel and consecutive perturbation as well as between the second and last perturbation. 

*Healthy older populations:* None of the included studies assessed changes in PEPs during repeated stance perturbations in healthy older adults.

*Pathological populations:* None of the included studies assessed changes in PEPs during repeated stance perturbations in pathological populations.

Task—Walking: None of the included studies assessed changes in PEPs during repeated gait perturbations in healthy young, healthy older and pathological populations.

Multiple tasks: None of the included studies assessed changes in PEPs during repeated perturbations across one or more different tasks in healthy young, healthy older and pathological populations.

#### 3.2.2. EEG Studies: Changes in Frequency Spectrums

One study in healthy young adults examined changes in frequency spectrums during repeated perturbations delivered in sitting [97] (Table 6). 

Task—Sitting: 

*Healthy young population: Unpredicted perturbations only:* A study by Shirazi and Huang [97] delivered repeated perturbations during mid-extension versus full-extension of leg while seated stepping via a motorized stepper. Changes in theta band synchronization and desynchronization were analyzed between early (first 33%) and late (last 33%) perturbed trials. The study findings showed greater theta band synchronizations in anterior cingulate cortex and supplementary motor area during full-leg extension as compared to perturbations delivered during mid-extension. This was followed by task-specific theta desynchronization in supplementary motor area during post-perturbation recovery (i.e., desynchronization occurred in left supplementary motor area for mid-extension type and right supplementary motor area for perturbations during full-extension. While novel perturbations observed greatest theta band synchronizations in anterior cingulate cortex and supplementary motor area, subsequent perturbations resulted in reduced theta band synchronizations across anterior cingulate cortex and supplementary motor area. 

*Healthy older population:* None of the included studies assessed changes in cortical frequencies during repeated seated perturbations in healthy older adults. 

*Pathological populations:* None of the included studies assessed changes in cortical frequencies during repeated seated perturbations in pathological populations. 

Task—Standing: None of the included studies assessed changes in cortical frequencies during repeated stance perturbations in healthy young, healthy older and pathological populations.

Task—Walking: None of the included studies assessed changes in cortical frequencies during repeated gait perturbations in healthy young, healthy older and pathological populations.

Multiple tasks: None of the included studies assessed changes in cortical frequencies during repeated perturbations across one or more tasks in healthy young, healthy older and pathological populations.

#### 3.2.3. fNIRS Studies: Changes in Hemodynamic Responses

One study in healthy young adults examined changes in frequency spectrums during repeated perturbations delivered in walking [51] (Table 7). 

Task—Sitting: None of the included studies assessed changes in oxygenated hemoglobin during repeated seated perturbations in healthy young, healthy older and pathological populations.

Task—Standing: None of the included studies assessed changes in oxygenated hemoglobin during repeated stance perturbations in healthy young, healthy older and pathological populations.

Task—Walking: A study reported perturbation phase-dependent changes in bilateral hemispheres with perturbations delivered via split-belt treadmill during walking [51] (Table 7). More specifically, pre-perturbation walking phase showed increased activation in bilateral dorsolateral prefrontal cortex whereas post-perturbation recovery period showed greater activations in ventrolateral and frontopolar prefrontal cortex. Repeated perturbations resulted in minimal changes in all prefrontal regions during early adaptation; however, late adaptation trials resulted in switching of activity from dorsolateral and frontopolar prefrontal to orbitofrontal cortex. 

Multiple tasks: None of the included studies assessed changes in oxygenated hemoglobin during repeated perturbations across one or more tasks in healthy young, healthy older and pathological populations.

### 3.3. Behavioral and Kinematic Correlates of Cortical Activity (RQ3)

Eight included studies that examined PEPs, three included studies that assessed changes in cortical frequency spectrums and one included study (total: twelve) that examined hemodynamic changes during unpredicted perturbations also investigated corresponding behavioral or biomechanical outcome measures.

#### 3.3.1. EEG Studies: PEPs

*Healthy young population: Unpredicted perturbations only:* Adkin et al. [41] reported that N1 amplitudes recorded at midline central areas were negatively correlated with reduced perceived balance confidence, reduced perceived postural stability and positively correlated fear of falling during high-ground (1.6 m above ground) unpredicted perturbations compared to ground level. However, there were no associations between N1 characteristics and perceived anxiety. On the other hand, Sibley and colleagues [87] reported no associations between changes in N1 amplitude at midline central areas and autonomic responses (i.e., electrodermal responses) during high-ground (3.2 m above ground) versus ground level unpredicted perturbations. Payne and Ting [44] tested the associations between N1 amplitude during varying magnitude unpredicted perturbations with balance performance assessed by the distance walked on a narrow beam. The study reported negative correlations between N1 amplitude and the distance travelled on the beam [44]. Another study by Payne and colleagues [96] examined associations between N1 characteristics and lower limb EMG activity (early and late postural responses) during repeated perturbations. During repeated perturbations, a reduction in N1 amplitude was positively correlated with reduction in early postural responses but not with late postural responses [96]. Finally, a study by Mierau et al. [94] delivered repeated stance perturbations and reported positive correlations between changes in N1 amplitude with changes in postural sway and ankle plantarflexor EMG activity.

*Healthy older population: Unpredicted perturbations only:* Duckrow and colleagues [63] stratified older adults into mobile and frail older adults based on short term physical performance battery-SPPB (tasks: standing balance, walking speed and sit-to-stand). In addition, this study examined behavioral measures like gait velocity, single foot stance time, functional base of support and standing balance performance during conflicting sensory inputs (sensory-organization testing). This study reported that older adults with reduced mobility and shorter time-to-step during sensory-organization testing exhibited delayed P1 and prolonged intervals between N1 and N2.

*Pathological populations: Unpredicted perturbations only:* Jacobs and colleagues [93] reported a negative correlation between P2 amplitude and kinematic and psychometric measures in people with and without chronic low back pain. Kinematic measures included center of mass displacement and psychological measures like pain perception (brief pain inventory short form), fear avoidance beliefs questionnaire and coping strategies questionnaire. On the other hand, Allexandre et al. [95] delivered unpredicted forward and backward perturbations via movable platforms and reported positive correlations between N1 amplitude and clinical balance performance (Berg balance scores). In addition, this study found negative correlations between N1 latency and center of pressure displacement. 

#### 3.3.2. EEG Studies: Changes in Frequency Spectrums

*Healthy young population: Unpredicted perturbations only:* Two studies have examined associations between beta frequency bands and clinical balance measures (Mini Balance Evaluation Systems Test and the distance travelled on the balance beam) [82,85]. Beta frequencies recorded after unpredicted perturbations especially during late phases (150–250 ms after perturbation onset) were negatively correlated with balance performance (Mini Balance Evaluation Systems Test scores and the distance travelled on the balance beam) [82,85]. 

*Healthy older population:* None of the included studies examined associations between cortical activity and behavioral/biomechanical outcomes in healthy older adults.

*Pathological populations:* One study reported negative correlation between theta frequency band with clinical balance performance (Berg balance scores) in individuals with traumatic brain injury and their healthy counterparts [83].

#### 3.3.3. EEG Studies: Changes in Hemodynamic Responses

*Healthy young population:* None of the fNIRS included studies examined the associations between cortical activations and behavioral/kinematic outcomes in healthy young adults.

*Healthy older population:* None of the fNIRS included studies examined the associations between cortical activations and behavioral/kinematic outcomes in healthy older adults.

*Pathological populations:* One fNIRS study reported positive correlation between changes in oxygenated hemoglobin in the supplementary motor area and prefrontal cortex after perturbation onset with clinical balance performance (Berg balance scale) in individuals with chronic subcortical stroke [65]. 

## 4. Discussion

We examined 35 articles that used mobile neuroimaging (EEG and fNIRS) to investigate mechanical perturbation-induced changes in cortical activity among healthy and clinical populations. Further, we examined 12 articles that investigated associations between cortical activity during reactive balance tasks and behavioral/biomechanical measures. Reactive balance control was most examined in healthy young during upright stance with cortical activity commonly assessed using PEPs and behavioral outcomes were commonly assessed using clinical balance measures (e.g., Berg balance scale, Beam Walking task). Comparatively lesser studies examined changes in cortical frequencies and hemodynamic responses during reactive balance tasks. In general, changes in cortical activity during reactive balance tasks were based on several internal (e.g., age or pathology) and external (e.g., task type: seated/stance/gait perturbations, task predictability, perturbation magnitude, concurrent cognitive task, task repetition). 

### 4.1. Spatial-Temporal Characteristics of Cortical Activity: Novel Perturbations (RQ1)

*Cortical involvement in feedback reception and sensorimotor processing:* Studies assessing modulations in PEPs and alpha and beta frequencies suggest the involvement of sensorimotor cortex in reception of peripheral sensory feedback and processing of perturbation characteristics. Moreover, age and pathology-related temporal changes in PEP P1, which is the first cortical event postulated to reflect afferent feedback reception, could be explained in two ways. First, delayed P1 in healthy older adults [63] could be attributed to slow conduction of peripheral afferents compared to young adults [99]. Second, delayed P1 in adults with traumatic brain injury [95] compared to young adults could arise from structural/functional alterations in the cortex itself [100]. Such delays in cortical perception of perturbation could have further resulted in alterations in sensorimotor processing [63,100], which is postulated to be reflected as the second cortical potential, PEP N1. Modulations in N1 were also seen in young adults. Specifically, changes in sensorimotor and parietal N1 responses were based on perturbation magnitude/postural threat [41,43,87], which could be explained in two ways. First, it is possible that with increasing perturbation magnitude/higher postural threat, young adults required greater cortical resources for sensorimotor processing. Alternatively, with higher perturbation magnitudes/greater postural threat there could be a greater mismatch between the expected and the ongoing postural state, thus resulting in higher cortical error-detection. On the other hand, there was reduction in N1 amplitudes with addition of a concurrent cognitive task [89,90,91,92] and during predicted perturbations [41,59,84]. This could suggest diversion of cognitive resources from ongoing processing in case of dual tasking or smaller mismatch between expected and ongoing state when there is prior knowledge/time to plan balance recovery in case of predicted perturbations. Remarkably, feet-in-place and compensatory stepping responses showed some similarities in sensorimotor frequency modulations [81], suggesting that the cortex might also have a role in feet-in-place responses. Specifically, both feet-in-place and compensatory stepping showed alpha reductions in occipital and supplementary motor areas during cue observation [81], suggesting alpha involvement in visual information processing or cortico-cortical information transfer. Further increased sensorimotor beta frequencies during perturbations with higher magnitudes [82] suggested their involvement in processing of perturbation characteristics or detecting changes in ongoing postural state. 

*Cortical involvement in planning and execution of reactive balance responses:* Changes in PEPs, beta and theta frequencies, and hemodynamic responses seen in prefrontal, frontal and sensorimotor areas suggested that the cortex is highly involved in planning and execution of reactive balance responses. Specifically, compensatory stepping responses elicited larger N1 amplitudes and increased beta frequencies in sensorimotor areas in young adults [43,81], especially during higher perturbation intensities [82], which could be explained in several ways. It is well-established that an effective compensatory step involves multiple components including timely initiation of step (i.e., foot liftoff, simultaneous shifting of body weight to the supporting stance limb and effective foot placement for balance recovery) [22,23]. In addition, executing a compensatory step might also depend on environmental conditions (e.g., floor height, space for stepping) and thus require greater attentional and visuospatial resources to plan appropriate responses. Thus, it can be postulated that compensatory stepping responses demand higher sensorimotor involvement for successful step execution. Contrastingly, when young adults were asked to execute feet-in-place responses at higher perturbation intensities, there were increased prefrontal theta frequencies compared to when asked to voluntarily take compensatory steps [81]. Hence, it is possible that cortical contribution to response execution is also based on task difficulty and not merely the type of balance response. This postulation can further be supported when reductions in frontal and sensorimotor P2 amplitudes were seen during seated compared to stance perturbations in young adults [42]. It is possible that small magnitude seated perturbations that also allow larger base of support might not be perceived with higher postural threat by the cerebral cortex. Secondly, it is possible that seated perturbations might require lesser cortical resources as they are comprised of trunk or single-joint movements for balance recovery compared to multi-segmented postural responses during stance perturbations. In older adults, increased prefrontal and motor beta frequencies were observed with lower compensatory stepping threshold [85], which suggested that aging adults might have heightened reliance on prefrontal cortex for step execution. Lastly, fNIRS studies also supported the involvement of prefrontal [50,51] and frontal [64] cortices in planning and execution of compensatory steps. Specifically, predicted perturbations induced greater hemodynamic changes in prefrontal, supplementary motor, and posterior parietal areas compared to unpredicted perturbations in young adults [64]. This suggested that prior knowledge of perturbations might involve greater recruitment of cognitive resources for pre-planning/decision making for the upcoming perturbation. In people with chronic stroke, cortical activations during balance recovery were more localized to the prefrontal cortex of the unaffected hemisphere [65], suggesting the pre-dominant role of unaffected hemisphere in planning of reactive balance responses.

### 4.2. Spatial-Temporal Characteristics of Cortical Activity: Repeated Perturbations (RQ2)

Included studies suggested reductions in the perturbation-evoked potential (PEP) N1 [72,94] and theta frequencies [97] with repeated perturbations in young adults. Further, there was switching of hemodynamic responses in prefrontal sub-regions with repeated perturbations [51]. It can be postulated that with repeated practice, there might be reduction in cortical error-detection and enhanced sensorimotor processing of perturbations resulting from recalibration of internal central nervous system (CNS) models and/or increased habituation of postural responses. Consequently, with repeated perturbations, there might be reduced reliance on the cerebral cortex resulting from more automated subcortical control based on stored motor memory or ongoing learning [101]. As all the included studies examined repeated perturbations in young adults, it remains unclear whether individuals aging and those with neurological impairments can demonstrate similar changes in cortical activity with repeated perturbations.

### 4.3. Behavioral and Biomechanical/Kinematic Correlates of Cortical Activity (RQ3)

A few included studies indicated direct associations between changes in cortical and biomechanical/kinematic correlates of reactive balance in young adults. Specifically, N1 reductions were associated with reductions in electromyographic responses [94,96] and postural sway during repeated perturbations [94]. It can be postulated that with repeated practice, there is enhanced sensorimotor processing resulting in smaller mismatch between the expected and ongoing postural state. This can result in smaller muscular activations required for balance recovery or improvement in balance performance.

Other studies indicated indirect associations between cortical activity changes during reactive balance tasks and behavioral measures assessed before/after the tasks. Specifically, changes in PEP N1 were associated with fear of falling and balance confidence [41,87], which suggested that perceived threat and psychological measures can interfere with cortical sensorimotor processing. Further, clinical dynamic balance performance was associated with changes in PEPs and cortical frequencies, as well as hemodynamic changes during reactive balance tasks [63,65,82,85,96]. This could suggest existence of shared pathways or common cortical areas of activation between volitional and reactive balance control. For example, previous neurophysiological literature indicated similar muscle synergy patterns between volitional and reactive stepping in healthy older adults [102] and associations between dynamic balance (Timed-up and go-test) and slip-related falls during walking in people with chronic stroke [103].

To summarize, even though the exact role of PEPs is not well established, their spatial-temporal changes provide a broad picture regarding the cortical control of reactive balance in healthy populations. Specifically, the role of later potentials P2 and N2 remains unknown possibly due to their undistinguishable nature from background EEG or from lack of advanced artifact removal techniques. Secondly, multicomponent PEPs provide an instantaneous snapshot of cortical changes related to reactive balance tasks; however, a majority of studies have used average trial analysis to quantify them. Previous neurophysiological studies have indicated that perturbation-induced motor adaptation can occur in as little as a single trial in healthy young and older adults [104,105,106]. Hence, future studies should test the feasibility and robustness of using single-trial PEP analysis. Additionally, assessing PEPs alone may not provide a clear picture of cortical modulation of reactive balance. Future studies could supplement PEP assessment with changes in alpha, beta or theta frequencies to provide a more accurate interpretation of changes in cortical activity during reactive balance tasks, especially for examining training (e.g., repeated perturbations) induced changes. Thirdly, there is still limited evidence regarding alterations in PEPs and cortical frequencies in aging and neurologically impaired populations (e.g., chronic stroke, multiple sclerosis). Further, majority of fNIRS studies used the prefrontal cortex as the area of interest. Hence, there is still lack of information regarding perturbation-induced whole-brain hemodynamic changes in aging and neurologically impaired populations. Lastly, it is worth highlighting that even though majority of included studies assessed behavioral/biomechanical measures, less than 35% of them examined their associations with cortical changes during reactive balance tasks. Further, none of the included studies established a true brain-behavior relationship for reactive balance control. Hence, it remains unclear whether changes in behavioral/biomechanical substrates are independent of or are driven by cortical modulation during reactive balance tasks among healthy and pathological populations.

## 5. Limitations of the Review

The current review was limited by several factors. Firstly, this review only included studies in adult populations (age > 18 years). Hence, this review lacked evidence from studies that examined physiological growth-related cortical changes in children and adolescents. Secondly, the keywords chosen for the search of research articles were based on current literature and individual preferences. Hence, the final research articles might have excluded studies with different terminology and emphasis that incorporated similar research paradigms. Thirdly, this study did not focus on providing evidence on movement and perturbation-related artifact removal techniques used in studies. Movement-related artifacts can restrict single-trial analyses of EEG signal data and limit evaluation of trial-to-trial changes in cortical activations during repeated perturbations. Lastly, while perturbation-related changes in PEPs, frequency spectrums, and hemodynamic changes provide information about cortical involvement in reactive balance control in healthy young, little is known about age and pathology-related alterations in brain activations during unpredicted perturbations. Thus, this review and existing literature lacks information regarding involvement of cerebral cortex in reactive balance control in high fall-risk populations (e.g., neurologically impaired populations). Lastly, the current review included research articles that were published only in English. It is possible that this review missed research articles in other languages that have published relevant findings related to cortical activations during reactive balance tasks using EEG and/or fNIRS. The authors would like to credit all the global centers that have conducted and/or published research regarding mobile neuroimaging of reactive balance control out of English science.

## 6. Conclusions

This systematic review demonstrated that, in general, unpredicted mechanical perturbations elicit cortical activations in prefrontal, sensorimotor, supplementary motor and posterior parietal areas. Specifically, the cerebral cortex might be involved for sensory processing of perturbations, error-detection of expected versus actual postural state, and planning and execution of compensatory stepping responses. Further, a handful of studies indicated that repeated exposure to perturbations could be associated with reduction in cortical error-detection and corresponding improvements in reactive balance performance. Lastly, task-related changes in cortical activity might be associated with biomechanical and clinical dynamic balance measures. As most of the current studies examined healthy young adults; future studies should focus on task-related cortical activations during novel and repeated perturbations and their behavioral correlates in aging and other populations with higher fall-risk. Future studies assessing PEPs could use single-trial analysis and supplement them with assessment of changes in frequency spectrums for examining cortical dynamics during reactive balance tasks. 

## Figures and Tables

**Figure 1 brainsci-12-01487-f001:**
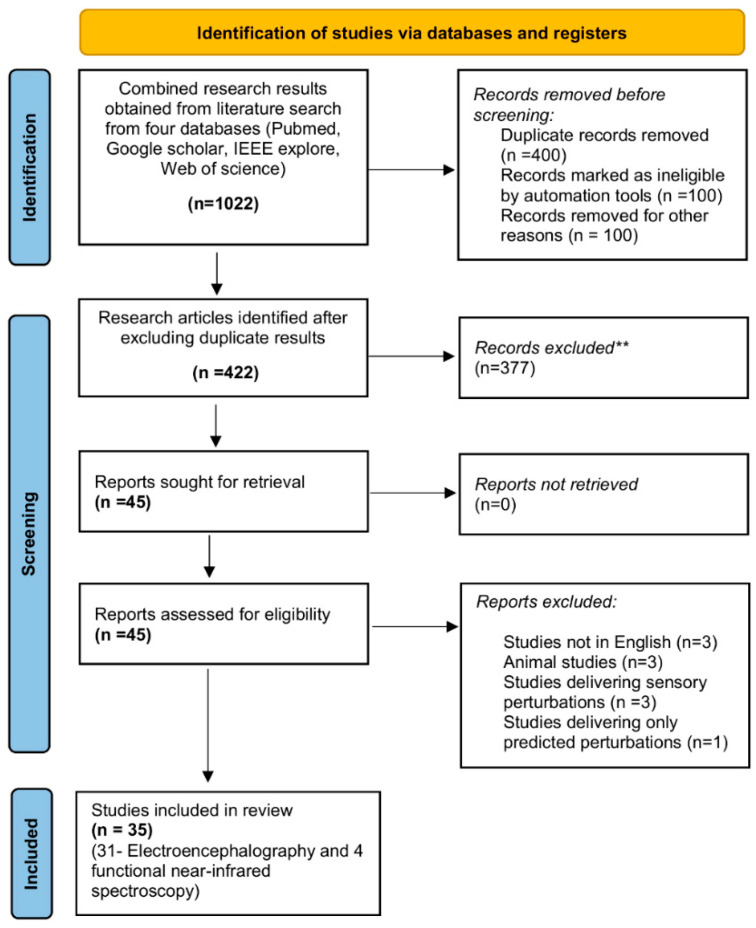
PRISMA 2020 flow diagram for new systematic reviews which included searches of databases: Combined search results from four databases yielded **1022** research articles out of which 600 were excluded due to duplication and ineligibility. The remaining **422** were assessed for eligibility and 377 were excluded after reviewing article abstracts. Thus, 10 articles were excluded from the remaining **45** after full-text analysis and finally **35** cross-sectional studies were included. ** Records excluded by the authors due to ineligibility.

**Table 1 brainsci-12-01487-t001:** Quality Assessment of Included Studies.

Study	Quality Assessment		Study	Quality Assessment	
	Author 1	Author 2		Author 1	Author 2
Ditz et al. [76]	M	M	Varghese et al. [77]	L	M
Goel et al. [78]	M	M	Varghese et al. [79]	L	M
Quant et al. [80]	M	M	Solis-Escalante et al. [81]	M	M
Payne and Ting [43]	M	H	Ghosn et al. [82]	L	M
Payne and Ting [44]	H	H	Solis-Escalante et al. [49]	M	M
Adkin et al. [59]	M	M	Shenoy et al. [83]	L	M
Mochizuki et al. [84]	L	L	Palmer et al. [85]	M	M
Adkin et al. [41]	M	M	An et al. [86]	M	M
Sibley et al. [87]	M	M	Mezzina et al. [88]	M	M
Bogost et al. [89]	M	M	Peterson and Ferris [48]	H	H
Little and Woollcott [90]	M	M	Mihara et al. [64]	L	L
Mochizuki et al. [91]	L	M	Mihara et al. [65]	L	L
Quant et al. [92]	M	M	Koren et al. [50]	M	M
Duckrow et al. [63]	M	M	Lee et al. [51]	M	M
Jacobs et al. [93]	M	M	Mierau et al. [94]	M	M
Alexandre et al. [95]	L	L	Payne et al. [96]	H	H
Dietz et al. [60]	L	L	Shirazi and Huang [97]	M	M
Mochizuki et al. [42]	L	L			

Abbreviations: H: High, M = Moderate, L = Low. Rating: ≥75% = High, 55–75% = Moderate, ≤55% = Low.

**Table 2 brainsci-12-01487-t002:** Perturbation-evoked potentials (PEPs) recorded via electroencephalography (EEG) during novel unpredictable (UNPRED) mechanical perturbations (RQ1).

Study	Sample	Type of Perturbations	Behavioral, Kinematic Outcomes	PEP	Electrode Sites	Key Findings
**Task position: Sitting**						
Ditz et al. [76]	15 young	UNPRED medial/lateral perturbations via mechanical chair	-	P1, N1, P2	Central, Parietal	N1 and P2 amplitude: UNPRED > rest No differences in P1
**Task position: Standing**						
Goel et al. [78]	10 young	UNPRED forward/backward, low/high perturbations via movable platform	-	N1	Frontal, Central	N1 amplitude: high > low for both N1 latency: high < low for both
Quant et al. [80]	7 young	UNPRED backward perturbations via movable platform (TASK 1: immediate deceleration TASK 2: delayed deceleration)	-	P1, N1, P2, N2	Central	All PEPs: TASK 1 ≈ TASK 2
Payne and Ting [43]	16 young	UNPRED backward perturbations (variable magnitude) via movable platform (planned/unplanned stepping, stepping/ non-stepping)	-	N1	Central	↑ N1 amplitude with ↑ perturbation magnitude, stepping > no-stepping, planned ≈ unplanned stepping
Payne and Ting [44]	20 young	UNPRED backward perturbations (3 levels: easy, moderate, difficult) via movable platform	Beam walking (balance)	N1	Central	↑ N1 amplitude with ↑ perturbation intensity ↓ N1 latency with ↑ perturbation magnitude N1 amplitude −ve correlation with Beam Walking task performance
Adkin et al. [59]	8 young	PRED and UNPRED perturbations forward perturbations via padded device	-	N1	Central	N1 latency: PRED ≈ UNPRED N1 amplitude: PRED < UNPRED
Mochizuki et al. [84]	15 young	PRED and UNPRED backward perturbations via load release	-	N1	Frontal, Central	Pre-perturbation activity: PRED > UNPRED N1 characteristics: UNPRED > PRED
Adkin et al. [41]	10 young	PRED and UNPRED forward perturbations at LOW (ground) and HIGH (3.2 m above ground) via hand-held bar	Balance confidence Fear of falling Perceived stability	N1	Central	N1 amplitude: HIGH > LOW for UNPRED, HIGH ≈ LOW for PRED N1 latency: same for all conditions Changes in N1 +ve correlation with changes in all behavioral outcomes
Sibley et al. [87]	10 young	UNPRED forward perturbations at LOW (ground) and HIGH (1.60 m above ground) via load release	Electrodermal responses	N1, P2	Central	N1 amplitude: HIGH > LOW, N1 latency, P2 amplitude: HIGH ≈ LOW No correlation between N1 and electrodermal responses
Bogost et al. [89]	15 young	UNPRED backward perturbations via movable platform with (DT) and without cognitive task (ST)	-	N1	Prefrontal, Premotor, Central, Centro- parietal	Regions of interest: ST: prefrontal, premotor, primary motor and supplementary motor area. DT: all areas for ST+ frontal, temporal and occipital area. N1 amplitude: ST > DT
Little and Woollcott [90]	14 young	UNPRED backward perturbations with cognitive task (DT) and cognitive task only (ST)	-	P1, N1 P2	Prefrontal, Premotor, Central, Centro- parietal	Regions of interest: ST: prefrontal, premotor, primary motor and supplementary motor area DT: all areas from ST+ temporal and occipital area. N1 amplitude: ST > DT
Mochizuki et al. [91]	26 young	PRED and UNPRED forward perturbations via lean release with (DT) and without cognitive task (ST)	-	N1	Central	Pre-perturbation activity: PRED: DT < ST N1 amplitude: PRED: ST < DT, UNPRED: ST ≈ DT
Quant et al. [92]	7 young	UNPRED forward/backward perturbations via moveable platform with (DT) and without (ST) cognitive task	-	N1	Central	N1 amplitude: DT < ST N1 latency: ST ≈ DT
Duckrow et al. [63]	8 young 32 old	UNPRED forward perturbations via movable platform	Subject height Physical performance battery	P1, N1, N2	Central	P1 latency: Old > Young N1 amplitude: Old < Young N2 latency: ↑ in Old with ↓ SPPB P1 latency +ve correlation with height N1-N2 interval −ve correlation with with physical performance
Jacobs et al. [93]	13 LBP 13 Healthy	UNPRED forward/backward perturbations via movable platform	Center of Mass displacement, Brief pain inventory, Fear avoidance Coping strategies	N1, P2	Frontal, Central, Centro- parietal	P2 amplitude: LBP > Healthy No group differences in N1 amplitude P2 amplitude −ve correlation with all behavioral outcomes
Alexandre et al. [95]	12 TBI 6 Healthy	UNPRED forward/backward perturbations via movable platform	Center of pressure displacement, Berg Balance scale	P1, N1, N2	Central	N1 amplitude: TBI < Healthy N1 amplitude +ve correlation with Berg Balance Scale
**Multiple task positions**						
Dietz et al. [60]	10 young	UNPRED forward perturbations in stance and gait via treadmill	-	P1, N1	Central	P1 latency: stance < gait N1 latency: stance < gait
Mochizuki et al. [42]	8 young	UNPRED backward perturbations in stance via load release and in sitting via chair tilting	-	N1, P2	Frontal, Central Centro- parietal	N1 characteristics at central (standing ≈ sitting) P2 amplitude: standing > sitting

Abbreviations: PRED: Predictable; UNPRED: Unpredictable; ↑: increase/enhanced; ↓: decrease/reduction; ≈: similar; >: greater than; <: less than; +ve: positive; −ve: negative; EMG: electromyography; TA: tibialis anterior, MG: Medial gastrocnemius, Central electrodes: Cz, C1, C2, C3; Frontal/Fronto-central: FPz, FP1, FP2, FP3, FP4; Centro-parietal: CPz, CP1, CP2, CP4; Parietal: Pz, P1, P2, P3, P4; LBP: Low-back pain; TBI: Traumatic brain injury.

**Table 3 brainsci-12-01487-t003:** Changes in frequency spectrums recorded via electroencephalography (EEG) in response to unpredictable (UNPRED) mechanical perturbations (RQ1).

Study	Sample	Type of Perturbations	Behavioral, Kinematic Outcomes	Frequency Band	Cortical Sites	Key Findings
**Task position: Standing**						
Varghese et al. [77]	14 Young	UNPRED forward perturbations via lean release	-	N1	Fronto- central	Simultaneous ↑ in power (theta, delta, alpha and beta) during N1
Varghese et al. [79]	14 Young	UNPRED forward perturbations via lean release	-	N1	N/A	Functional connectivity strength ↑ delta, theta, alpha and beta during N1
Solis-Escalante et al. [81]	10 Young	UNPRED backward perturbations via movable platform Phases: cue observation, response preparation, response execution (RE)	-	alpha, beta, theta	Supplementary motor, Sensorimotor, Prefrontal, Posterior parietal, Anterior cingulate cortex	ERSPs Cue observation: ↓ alpha + ↓ beta Response preparation: ↓ beta, ↓ theta, ↓ alpha and gamma Response execution: ↑ theta, alpha, beta. Stepping: ↓ beta in M1/S1 contralateral to support leg
Ghosn et al. [82]	19 Young	UNPRED backward perturbations (easy-difficult) via movable platform	Beam walking task (balance)	beta	Central	↑ beta power with ↑ perturbation magnitude Late phase beta power (150–250 ms post-perturbation) −ve correlation with balance performance
Solis-Escalante et al. [49]	3 Stroke 6 Young	UNPRED forward/backward, medial/lateral perturbations via movable platform	-	theta	Central	≈ pattern of changes in theta frequencies in both groups (i.e., ↑ theta after perturbation onset)
Shenoy et al. [83]	18 TBI 18 Healthy	UNPRED anterior/posterior perturbations with low/high amplitude via movable platform	Center of pressure displacement, Berg Balance Scale	alpha beta theta	Frontal, Parietal, Temporal, Occipital	Regions-of-interest alpha connectivity: TBI < Healthy, beta connectivity: TBI ≈ Healthy in ROIs Theta band modularity −ve correlation with Berg Balance Scale, no correlation with center of pressure displacement
Palmer et al. [85]	16 Old	UNPRED forward/backward, medial/lateral perturbations via movable platform	Mini Balance evaluation test (balance), Cognitive motor interference, Stepping threshold	beta	Primary motor, Sensory, Prefrontal	↑ prefrontal, primary motor beta connectivity post-perturbation correlated with ↓ stepping threshold Beta power during late phase (150–250 ms) −ve correlation with MiniBEST.
**Task position: Walking**						
An et al. [86]	5 Young	UNPRED backward perturbations via split-belt treadmill Phases: Quiet standing, walking, recovery response.	-	alpha beta theta delta gamma	Sensorimotor, Posterior parietal	In sensorimotor cortex, ↑ theta power: walking > standing, ↑ alpha power: recovery < standing/walking In posterior-parietal cortex, ↑ theta power: walking > standing, ↑ alpha power: recovery > standing/ walking, ↑ beta power: recovery/walking < standing
Mezzina et al. [88]	4 Young	UNPRED forward perturbations via split-belt treadmill	-	alpha beta theta	Frontal, Parietal	Cortical responsiveness (*slope m*) ↑ in m of all bands post perturbation. ↓ in m during recovery > walking and early balance loss phase
**Multiple task positions**						
Peterson and Ferris [48]	30 Young	UNPRED perturbations in standing or walking on a balance beam via waist pull Conditions: stand pull, walk pull	-	alpha beta theta and gamma	Occipital, Posterior parietal, Sensorimotor, Supplementary motor area	↑ alpha, ↑ beta power: stand pull > walk pull in sensorimotor, posterior parietal and supplementary motor area ↓ gamma power: stand pull < walk pull in occipital and posterior parietal area

Abbreviations: UNPRED: Unpredictable; ↑: increase/enhanced; ↓: decrease/reduction; ≈: similar or comparable; >: greater than; <: less than; central: Cz, C1, C2, C3, C4; Primary motor: Cz, primary sensory: CPz; Prefrontal: AFz; Frontal: F3, FZ, F4; Centro-parietal: CP1, CP2, CP6; ERSP: Event-related spectral perturbations; TBI: Traumatic brain injury; M1: Motor cortex, S1: Sensory cortex.

**Table 4 brainsci-12-01487-t004:** Changes in oxygenated and deoxygenated hemoglobin (Hb) levels recorded via functional near-infrared spectroscopy (fNIRS) in response to novel and repeated unpredictable (UNPRED) mechanical perturbations (RQ1).

Study	Sample	Type of Perturbations	Behavioral, Kinematic Outcomes	Cortical Responses	Cortical Area	Key Findings
**Task position: Standing**						
Mihara et al. [64]	15 Young	UNPRED and PRED forward/backward, medial/lateral perturbations via movable platform	-	OxyHb, Deoxy Hb	Frontal, Parietal, Primary motor	↑ OxyHb: PRED, UNPRED > pre-perturbation in frontal and parietal ↑ OxyHb: PRED > UNPRED in superior parietal & supplementary motor area
Mihara et al. [65]	20 Stroke	UNPRED forward/backward perturbations via movable platform	Berg Balance Scale (balance)	OxyHb, Deoxy Hb	Prefrontal Premotor Parietal	↑ OxyHb: post > pre-perturbation in prefrontal, and parietal of unaffected hemisphere ↑ OxyHb in supplementary motor area and prefrontal cortex +ve correlation with Berg Balance scale
**Task position: Walking**						
Koren et al. [50]	20 Young	UNPRED over ground walk perturbations via mechatronic system. 3 conditions: Unperturbed walk, perturbed walk and own shoes	-	OxyHb, Deoxy Hb	Prefrontal cortex	↑ OxyHb: perturbed walk > unperturbed/shoe in prefrontal cortex
Lee et al. [51]	10 Young	UNPRED repeated slips via split-belt treadmill	-	OxyHb, Deoxy Hb	Prefrontal cortex and sub regions	↑ OxyHb: walking, recovery period > pre-perturbation in prefrontal cortex

Abbreviations: PRED: Predictable; UNPRED: Unpredictable; OxyHb: Oxygenated Hemoglobin, DeOxyHb: Deoxygenated Hemoglobin; ↓: reduced; ↑: increased.

**Table 5 brainsci-12-01487-t005:** Perturbation-evoked potentials (PEPs) recorded via electroencephalography (EEG) in response to repeated unpredictable (UNPRED) mechanical perturbations (RQ2).

Study	Sample	Type of Perturbations	Behavioral, Kinematic Outcomes	PEP	Electrode Sites	Key Findings
** Task position: Standing **						
Mierau et al. [94]	37 Young	Repeated UNPRED forward/backward, medial/lateral perturbations via movable platform (10 trials: T1-T10)	Postural sway Electromyographic responses	P1, N1	Fronto-central Centro-parietal	N1 amplitude: T1 > T2 > T10 N1 latency: T1 ≈ T2 ≈ T10 P1 characteristics: T1 ≈ T2 ≈ T10 N1 amplitude +ve correlation with postural sway and electromyographic responses
Payne et al. [96]	16 Young	Repeated UNPRED forward/backward perturbations via movable platform	Electromyographic responses, (early and late)	N1	Central	↓ N1 amplitude with ↑ number of trials N1 amplitudes +ve correlation with early but no correlation with late electromyographic responses

Abbreviations: UNPRED: Unpredictable; ↑: increase/enhanced; ↓: decrease/reduction; ≈: similar; >: greater than; <: less than; EMG: electromyography; TA: tibialis anterior, MG: Medial gastrocnemius, LG: Lateral gastrocnemius; Central electrodes: Cz, C1, C2, C3; Frontal/Fronto-central: FPz, FP1, FP2, FP3, FP4; Centro-parietal: CPz, CP1, CP2, CP4; Parietal: Pz, P1, P2, P3, P4.

**Table 6 brainsci-12-01487-t006:** Changes in frequency spectrums recorded via electroencephalography (EEG) in response to repeated unpredictable (UNPRED) mechanical perturbations (RQ2).

Study	Sample	Type of Perturbations	Behavioral Kinematic Outcomes	Frequency Band	Cortical Sites	Key Findings
**Task position: Sitting**						
Shirazi and Huang [97]	17 young	Repeated UNPRED perturbations in sitting during mid-leg or full-leg extension onset via servo meter	-	theta	Supplementary motor area, Anterior cingulate cortex	↑ theta power: pre-> post-perturbation in Right supplementary motor area ↑ theta power: post > pre-perturbation in anterior cingulate and Left supplementary motor area ↓ theta power with repeated perturbations

Abbreviations: UNPRED: Unpredictable; ↓: reduced; ↑: increased.

**Table 7 brainsci-12-01487-t007:** Changes in oxygenated and deoxygenated hemoglobin (Hb) levels recorded via functional near-infrared spectroscopy (fNIRS) in response to repeat-ed unpredictable (UNPRED) mechanical perturbations (RQ2).

Study	Sample	Type of Perturbations	Behavioral Kinematic Outcomes	Frequency Band	Cortical Sites	Key Findings
**Task position: Walking**						
Lee et al. [51]	10 young	Repeated UNPRED backward perturbations during walking via split-belt treadmill	-	OxyHb and Deoxy Hb	Prefrontal cortex and subregions	Trial 1–3: ≈ OxyHb in prefrontal sub-regions Trial 3–6: ↓ OxyHb in ventrolateral and frontopolar prefrontal cortex, ↑ OxyHb in orbitofrontal cortex.

Abbreviations: PRED: Predictable; UNPRED: Unpredictable; OxyHb: Oxygenated Hemoglobin, De-OxyHb: Deoxygenated Hemoglobin; ↓: reduced; ↑: in-creased.

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
