# Peer review of "Mobile Brain Imaging to Examine Task-Related Cortical Correlates of Reactive Balance: A Systematic Review"

_brainsci, 2022, doi:10.3390/brainsci12111487_

Round 1

Reviewer 1 Report

This review examined available findings on spatial and temporal characteristics of cortical activity in response to unpredicted mechanical perturbations, and investigated associations between cortical activity and behavioral/biomechanical measures in adult population. This review is well articulated and clearly explained. However, there are several areas needed to be considered before publication. I hope the authors could further strengthen the methodological and discussion section regarding the study findings. Here are my key concerns:

·        Was this systematic review protocol registered in the Prospective International Register of Systematic Reviews, PROSPERO?

·        The methodological quality of the randomized studies could be evaluated by two reviewers.

·        The order in the methodology could be as follows: search strategy, eligibility criteria, quality assessment, data extraction, data analysis. In addition, the data from the PEDro scale would be included in the section quality assessment.

·        Were the search terms combined to perform a more comprehensive search? If so, please indicate the combinations made.

·        In the data extraction section, please include information about the studies selected. For example: Specific information was extracted for the systematic review from the studies selected, which included: (1) first author and year of publication; (2) sample size of the study; (3) characteristics of the sample; (4) characteristics of the disease; ….

·        The introduction section is quite long. The list of previous evidences could be summarized and reorganized to better highlight what this review adds to the existing literature.

·        The systematic review is correct and exhaustive, I think it is a great work that would benefit greatly from complementing it with a meta-analysis.

·        The discussion resembles in part the results, it would better integrate the concepts found. In the results a great job is done in dividing the findings according to the technique and the questions of interest, in the discussion a better complementation of the concepts and contents is expected, not a mere description of the results again.

Reviewer 2 Report

Review report-1889050-brainsci

A brief summary

Goal of the review was to provide evidence regarding the involvement of cerebral cortex for sensory processing of unpredicted perturbations, error-detection of expected versus actual postural state, planning and execution of compensatory stepping responses. According to authors, study demonstrated that, in general, reactive balance tasks elicit cortical activations in prefrontal, sensorimotor, supplementary motor and posterior parietal areas. Specifically, analyses revealed that the cerebral cortex might be involved for sensory processing of perturbations, error-detection of expected versus actual postural state as well as execution of compensatory stepping responses.

Broad comments

Introducing overview was supported by methodology used to fulfill aim of the study. Review design was genneraly appropriate, followed and supported by relevant and precise conclusions. Limitations of the study were mostly correctly explained. However, those should be extended by additional points mentioned in Specific comments. Finnaly, study presenta itself as a very important contribution for specific and interdisciplinarity-oriented practice of researcher, clinitian or a team.

Specific comments.

·       Minor spell check, and style corrections (e.g. 22,42,43,50 and further on) become frustrating major irritation for reader

·       Figure 1. ‘Studies not in English’ (it has to be set within important limitation; on the other hand, It would not be professional from a reviewer to point out research within one or two local national ‘brain institutes’, whereas relevant findings were published in French, German, Croatian, Russian or likewise languages) – at the same moment it is an advantage and huge disadvantage for this field of study (like in math – universal language) - please give credit to global centers of relevance within and out of ‘english’ science by short mention within limitation paragraph or one sentence

·       All abbreviations, e.g.  ST, DT and other – need description somewhere – in text, under table…

·       In table 2., within Key Findings for Ghosn et al [85] info not clearly written (check all content within tables 1-6)

·       With regard to title, biomechanics (kinematic, sEMG, kinetic) parameters should have been a criterion set as much more important for the review (with regard to inclusion/exclusion criteria and 3 major ‘questions’) – mentioned but importance not stressed sufficiently (even thou biomechanics is a ‘quantification’ of motor control…)  Ln 250-2: “Although only 12 studies examined the associations between cortical activations and behavioral/biomechanical outcomes [46, 49, 67, 69, 80, 85, 87, 90, 93, 94, 97, 99], a majority of the included studies assessed behavioral/biomechanical measures”; likewise repeated In further text

·       scrutiny from authors over statistical power and sample size calculations within mentioned research should also have been introduced – not all study fully applied all st.power criterion…
